# Inoculation of *Klebsiella variicola* Alleviated Salt Stress and Improved Growth and Nutrients in Wheat and Maize

Supriya P. Kusale [1], Yasmin C. Attar [1,*], R. Z. Sayyed [2], Hesham El Enshasy [3,4,*], Siti Zulaiha Hanapi [3], Noshin Ilyas [5], Abdallah M. Elgorban [6], Ali H. Bahkali [6] and Najat Marraiki [6]

1 Department of Microbiology, Rajaram College, Kolhapur 416004, India; supriya.kusale@gmail.com
2 Department of Microbiology, P.S.G.V.P. Mandal's, Arts, Science, and Commerce College, Shahada 425409, India; sayyedrz@gmail.com
3 Institute of Bioproduct Development (IBD), Universiti Teknologi Malaysia (UTM), Skudai, Johor Bahru 81310, Malaysia; zulaiha@ibd.utm.my
4 City of Scientific Research and Technology Applications, New Burg Al-Arab, Alexandria 21934, Egypt
5 Department of Botany, PMAS Arid Agriculture University, Rawalpindi 46300, Pakistan; noshinilyas@yahoo.com
6 Department of Botany and Microbiology, College of Science, King Saud University, P.O. Box 2455, Riyadh 11451, Saudi Arabia; aelgorban@ksu.edu.sa (A.M.E.); abahkali@ksu.edu.sa (A.H.B.); najat@ksu.edu.sa (N.M.)
* Correspondence: yasmin.attar33@gmail.com (Y.C.A.); henshasy@ibd.utm.my (H.E.E.)

**Abstract:** Although wheat and maize are the major economically important cereal crops and staple food sources in the world, their productivity is highly affected by excess salts in soil (salinity). Applications of multifarious halophilic plant growth-promoting rhizobacteria (PGPR) in saline soil protect the plants from osmotic damages and promote plant growth through the secretion of plant growth promoting (PGP) and osmolytes. In this study, *Klebsiella variicola* SURYA6—a PGPR—was evaluated for plant-growth-promotion and salinity amelioration in wheat and maize, and enrichment of soil nutrients. The results of the present study revealed that *K. variicola* SURYA6 grows luxuriously under high salinity stress conditions and produces copious amounts of three principal salinity ameliorating traits, such as 1 aminocyclopropane-1-carboxylate deaminase (ACCD), indole-3-acetic acid (IAA), exopolysaccharides (EPS), and osmolytes—such as proline, sugars, proteins, and amino acids. The isolate also exhibited sensitivity to a wide range of antibiotics, lack of hemolytic ability, and absence of catalase and oxidase activities confirming its nonpathogenic nature. Inoculation of wheat and maize seeds with this multifarious strain, improved the physicochemical properties of soil, improved seed germination by 33.9% and 36.0%, root length by 111.0%, 35.1%, shoot height by 64.8% and 78.9%, and chlorophyll content by 68.4% and 66.7% in wheat and maize seedlings, respectively, at 45 days after sowing (DAS) under salinity stress. The improvement in plant growth can be correlated with the secretion of PGP traits and improved, uptake of minerals such as nitrogen (N), phosphorus (P), sodium (Na), potassium (K), and magnesium (Mg). While amelioration of salinity can be the result of secretion of osmolytes and the change in pH from salinity to neutrality. This inoculation also significantly improved the soil nutrients under salinity stress conditions. Inoculation of *K. variicola* SURYA6, resulted in more improved growth and nutrients contents in plants and enriched soil nutrients under salinity stress as compared to normal (non-saline) conditions. Such multifarious strain can serve as a potent bio-inoculant for growth promotion of wheat and maize in saline soil. However, multi-year field trials under different agro-climatic conditions are required to confirm the bio-efficacy of *K. variicola* SURYA6.

**Keywords:** maize; osmolytes; plant growth promotion; salt stress; soil nutrients; wheat

## 1. Introduction

Wheat (*Triticum aestivum* L.) is one of the most important cereal crops for the majority of people around the world [1]. It is the second most-produced cereal after maize [2].

However, the present rate of increase in population predicted that by 2050 consumers will require 60 percent more wheat than today [3]. While maize (*Zea mays* L.) is one of the most versatile crops also known as the queen of the cereals [4]. However, it is sensitive to salinity stress [5]. Although these two crops are the major, widely cultivated, and economically important crops, their productivity is severely affected by abiotic stresses, such as drought and salinity [5].

The presence of excess salts in the agriculture fields is an increasing problem worldwide [6]. It is estimated that about 27% —i.e., one-third of the world's arable land is already affected by salinity [7]. Saline soil prevents the normal growth of crops and results in poor crop yield or sometimes total failure of the crop [8,9]. The conventional methods of amelioration of soil salinity—i.e., the use of salt-tolerant crops [10], plant breeding, soil scrapping, chemical leaching of excess salts by adding, gypsum, calcium chloride, etc.— have provided some success, but pose negative impacts on the soil health [11]. Moreover, these approaches are beyond the reach of developing nations [12]. This warrants the urgent need for sustainable approaches for improving the crop productivity in saline soil and ameliorating the soil salinity without compromising the soil health [13]. In this regards the application of plant growth-promoting rhizobacteria (PGPR) has proved useful to promote plant growth in saline soils [14,15]. Many PGPR such as *Rhizobium* sp. *Bradyrhizobium* sp., *Azotobacter* sp., *Azospirillum* sp., *Bacillus* sp., *Alcaligenes* sp., *Enterobacter* sp., *Pseudomonas* sp., *Klebsiella* sp., and other halotolerant PGPR protects the plant from damages caused by salinity and promotes plant growth as well [15]. PGPR imparts a spectrum of beneficial effects to crop plants including the amelioration of soil salinity [16]. Under salt stress, various PGPR including *Enterobacter* sp. [15], *E. cloacae* KP226569 [16], SU18 *Arthrobacter* sp. [17] produces phytohormones such as indoleacetic acid (IAA) [18]; exopolysaccharides (EPS) [19]; osmolytes like proline, sugars, amino acids, proteins [19], and aminocyclopropane-1-carboxylate deaminase (ACCD) [8,15] and known to relieve the negative impacts of excess salts in rice and wheat [20]. During salinity, PGPR produces EPS that chelates excess $Na^+$ of salt and reduce its level [21]. ACCD positive PGPR hydrolyzes the ethylene precursor (ACC) and reduces the level of ethylene during excess salt conditions [21,22]. Decreased levels of ethylene help in the good growth of roots and improved absorption of nutrients.

*Klebsiella variicola* SURYA6 is associated with plants and clinical samples [23]. Although it occurs in clinical samples and animals, its plant-growth-promoting and salinity ameliorating potentials are not well studied [23]. Moreover, the reports on its plant growth-promoting ability under normal soil and saline soil are scarce. This warrants the need to explore the plant-growth-promoting and salinity ameliorating potential of this PGPR strain. The present study was aimed to evaluate the plant growth promoting and ability of in wheat and maize under salinity stress, and its ability to enrich the soil nutrient contents under saline conditions.

## 2. Materials and Methods

### 2.1. Soil Samples, Plant Seeds, and Klebsiella variicola SURYA6

A one kg of rhizosphere soil from each of four four different corners of wheat and maize fields were collected from Nandani (16.7262° N, 74.5434° E) village of Kolhapur District of Maharashtra, India. This soil was used for sowing bacterized seeds and for evaluating the plant-growth promoting effects of *K. variicola* SURYA6, as this organism was previously isolated from this local soil sample. Seeds of wheat variety Phule Samadhan NIAW-1994, and maize, variety Rajarshi-hybrid KMH-22168 used in the present study and were procured from a local market in Kolhapur, Maharashta, India. *K. variicola* SURYA6 used in this study was obtained from the Department of Microbiology, Rajaram College, Kolhapur, Maharashtra, India. Since the isolate was obtained from wheat and maize rhizosphere and it produced multiple plant growth-promoting (PGP) traits [24], it was checked for its ability to ameliorate soil salinity, promote growth of wheat and maize under

normal and saline soil, (prepared by adding 100 mM of NaCl per kg of soil) and improve osmolyte and nutrients in plants and nutrients in the soil.

### 2.2. Analysis of Physicochemical Parameters of the Soil before and after Sowing

The physical and chemical properties of soil were analyzed before and 45 days after sowing (DAS). The pH and electrical conductivity were measured by pH and electrical conductivity meters, respectively. For the estimation of the available nitrogen of the soil of control and test pots, 1 g of dried leaf biomass was digested in 10 mL concentrated $H_2SO_4$ and 5 g catalyst mixture in a digestion tube. The digested and cooled mixture was distilled and the distillate was titrated with $H_2SO_4$. Mixture that did not contain leaf extract served as a control. Total nitrogen was calculated from the blank and sample titer reading [25]. The P content of soil was first extracted with 0.5 N $NaHCO_3$ buffer (pH 8.5) followed by treatment with ascorbic acid and the intensity of blue color produced was measured at 520 nm. The amount of P from soil was calculated from the standard curve of P [26]. For the estimation of potassium content of soil, one g of soil was added in 25 mL of ammonium acetate, shaken for 5 min and filtered. The amount of potassium from the filtrate was measured according to the method of Baghel [27]. For the estimation of Ca, Mg, K, Fe, Zn, Cu, Mn, and B, 1 g of soil was mixed with 80 mL of 0.5 N HC1 and incubated for 5 min at 25 °C followed by filtration. The amount of these minerals in the filtrate was estimated according to the method of Gupta and Mohopatra [28].

### 2.3. Evaluation of Salinity Tolerance in K. variicola SURYA6

*K. variicola* SURYA6 was screened for its potential to grow over the range (0–200 mM) of various salts such as NaCl, KCl, $Na_2CO_3$, and $Na_2HCO_3$ in nutrient broth (NB). A log phase culture ($5 \times 10^{-6}$ cells/mL) of *K. variicola* SURYA6 was grown in each NB flask separately amended with each concentration of a particular salt. Inoculated NB flasks were incubated at 28 °C at 120 rpm for 24–48 h. After incubation, the optical density (OD) for growth of the organism from each flask was measured spectrophotometrically at 620 nm [8].

### 2.4. Screening for Salinity Ameliorating Metabolites of K. variicola SURYA6

#### 2.4.1. Production of ACCD

ACCD activity of *K. variicola* SURYA6 was screened in minimal medium (MM) containing 2, 0.5, 0.2, 0.2, and 0.19 g/L of $KH_2PO_4$, $K_2HPO_4$, $MgSO_4$, glucose, and $(NH_4)_2SO_4$, respectively. A log phase culture ($5 \times 10^{-5}$ cells/mL) of *K. variicola* SURYA6 was grown on MM agar and in MM broth at 28 °C for 48 h. Following the incubation, plates were observed for the appearance of the growth of the organism [29,30]. MM broth was centrifuged at 10,000 rpm for 10 min and the supernatant was used to estimate ACCD activity. ACCD activity was defined as the amount of α-keto-butyrate produced per mg of protein per h [31].

#### 2.4.2. Screening and Production of Indole-3-Acetic Acid (IAA)

For screening of IAA production, log phase culture of *K. variicola* SURYA6 ($5 \times 10^{-5}$ cell/mL) was grown in tryptophan (0.2%) containing NB at 28 ± 2 °C for 48 h at 120 rpm. Following the incubation, broth was centrifuged at 10,000 rpm for 20 min and in 2 mL of supernatant, 2 drops of orthophosphoric acid, 4 mL of the Salkowski's reagent and 1 mL of 0.5 M $FeCl_3$ solution were added. This reaction mixture was incubated at 28 °C for 30 min and the intensity of red color was measured spectrophotometrically at 530 nm [32]. The uninoculated broth served as a reference and the amount of IAA was calculated from the calibration curve of IAA.

#### 2.4.3. Screening and Production of Exopolysaccharide (EPS)

The ability of *K. variicola* SURYA6 to produce EPS was screened in a 5% glucose basal medium (BM). A log phase culture ($5 \times 10^{-5}$ cells/mL) of the organism was grown in BM at 28 ± 2 °C for 72 h at 120 rpm. After the incubation, the BM was observed for an increase in the viscosity as an indication of EPS production. The viscosity of inoculated BM

was measured with viscometer at 29 °C at 100 rpm and was expressed as centipoises (cP). Uninoculated BM was used as a reference medium. For quantitative estimation of EPS, BM was centrifuged at 10,000 rpm for 20 min. EPS was precipitated and extracted with cold ethanol followed by drying in an oven at 50 °C until the constant weight was achieved. The amount of EPS was measured gravimetrically as g/L dry weight [33,34].

### 2.5. Confirmation of the Non-Pathogenic Nature of the Potent Isolate

To confirm the non-pathogenic nature of *K. variicola* SURYA6 for humans, antibiotic sensitivity, hemolysis, catalase, and oxidase tests were performed.

### 2.5.1. Antibiotic Sensitivity and Blood Agar Test

The sensitivity of *K. variicola* SURYA6 to different antibiotics was tested according to the method of Bauer et al. [35]. For this purpose, the log phase culture ($5 \times 10^{-5}$ cells/mL) of *K. variicola* SURYA6 was spread on the Muller Hinton (MH) media, and antibiotic discs were placed on the surface of the plate. Plates were incubated at $37 \pm 2$ °C for 24–48 h and after the incubation plates were observed for the sensitivity or resistivity in terms of growth or inhibition of the growth of the isolate on MH medium.

### 2.5.2. Hemolysis Test

The confirmation of the non-hemolytic nature of the *K. variicola* SURYA6 was carried out by growing log phase culture ($5 \times 10^{-5}$ cells/mL) of this organism on BA (nutrient agar with 5% sheep RBC) at $37 \pm 2$ °C for 24–48 h [36]. Following the incubation, plates were observed for the presence or absence of zone of hemolysis.

### 2.5.3. Catalase and Oxidase Tests

For the catalase test, a loop full of log phase culture of *K. varricola* SURYA6 was added in 3% ($v/v$) hydrogen peroxide ($H_2O_2$) solution and observed for the effervescence of oxygen as an indication of the breakdown of $H_2O_2$ due to the production of catalase [37].

For oxidase test, the wet filter paper method was used [38]. For this purpose, a strip of Whatman's filter paper was soaked in a freshly prepared 1% ($w/v$) solution of N,N,N′,N′-tetramethyl-p-phenylenediamine dihydrochloride (TMPD) dye. The excess dye was allowed to drain out from the strip for about 30 s. The colony of log phase culture of *K. varricola* SURYA6 was smeared over the dye strip and observed for the development of intense deep-purple hue, appearing within 5–10 s as an indication of oxidase production.

### 2.6. Plant Growth Promotion Studies
#### 2.6.1. Experimental Design

This research was carried out from August through September, 2019 in the pots under greenhouse conditions. Pots of 1.5 kg capacity containing 1.0 kg of sandy loamy soil, i.e., the same soil from which the samples were collected. Pots were separately sown with bacterized seeds of wheat and maize (5 seeds/pot), non-bacterized seeds served as control. Seeded pots were kept under greenhouse conditions at 28 °C with 65–70% relative humidity and a 12–14 h photoperiod. The pot assay was performed in completely randomized design (CRD) with four treatments and five replications each for wheat and maize seeds. The treatments were as follows T0 = Non-bacterized seeds sown in normal soil. T1 = Bacterized seeds sown in normal soil. T2 = Non-bacterized seeds sown in saline soil. T3 = Bacterized seeds sown in saline soil.

Seedlings were uprooted after 30 and 45 days of sowing and subjected to the measurement of growth, nutrient, and osmolyte contents in plants and nutrient contents in soil.

#### 2.6.2. Making of Saline Soil

The sandy loamy soil from the agriculture fields of Kolhapur, Maharashtra, India was collected and subjected to the measurement of various soil parameters. Salt (NaCl) was added to this soil at the rate of 100 mM/kg. This concentration was chosen based on the

average amount of NaCl present in saline soil which varies from 48 to 111 mM [8,9]. At the end of the experiment, the observed level of NaCl was 100 mM NaCl/Kg soil

### 2.6.3. Seed Bacterization and Application of *K. variicola* SURYA6 to the Soil

*K. variicola* SURYA6 was grown in NB at $28 \pm 2$ °C for 24–48 h at 120 rpm. After 24 h incubation, the log phase culture of *K. variicola* SURYA6 ($5 \times 10^{-6}$ CFU/g) was used for the bacterization of wheat and maize seeds. Seeds of wheat and maize were surface sterilized with 0.1% $HgCl_2$ and 70% ethanol treatment for 1 min followed by three washing with sterile distilled water. For bacterization, surface-sterilized seeds were immersed in the culture broth of *K. variicola* SURYA6 for 20 min, dried in shade, and sown in pots (5 seeds/pot) as per the above treatment. Control (non-bacterized) seeds were immersed in uninoculated (free from *K. variicola* SURYA6) NB. Pots were placed in the greenhouse and watered daily for 30 and 45 days. Normal water for normal soil and saline water (50 mM NaCl) for saline soil was used.

### 2.6.4. Measurement of Plant Growth Parameters

For the measurement of plant growth parameters, five seedlings each of wheat and maize were uprooted from the soil after 30 and 45 days and subjected to the measurement of seed germination, root length, shoot height, shoot dry weight, the number of leaves, and chlorophyll content. Seed germination was calculated in percent and transformed in arcsine values, root, and shoot height were measured in cm, and shoot weight was measured on a dry weight basis. For chlorophyll estimation, fresh leaves of test and control plants were collected at 30 and 45 DAS, washed thoroughly, and 1 g of leaf tissue was crushed in 80% ($v/v$) acetone followed by centrifugation at 10,000 rpm for 5 min and filtration through Whatman filter paper No 1. The absorbance of the supernatant was read at 663 and 645 nm and the total chlorophyll content was determined according to the method of Arnon [39].

### 2.6.5. Estimation of Osmolyte, Sugar, Protein, and Amino Acid Contents in Plants

Osmolytes such as proline, soluble sugars, proteins, and amino acids, were estimated from control and test plants. One g of seedling material of each treatment plant was separately crushed in 100 mM phosphate buffer (pH 7) and 80% ethanol, respectively. Proline content from leaf extract of each treatment plant was estimated according to the method of Bates [40]. For this, one mL of the leaf filter extract was mixed in one mL of ninhydrin reagent (5% ninhydrin in glacial acetic acid: phosphoric acid (100:80 $v/v$) and incubated in a water bath at 90 °C for 1 h and cooled in ice water. This was followed by the addition of 1 mL of toluene and the proline content from the resulting upper layer of the reaction mixture was measured at 520 nm. The amount of proline content was calculated from the standard curve of proline. The soluble sugar content was determined according to the method of Dubois et al. [41]. The leaf extract was mixed with 3 mL of 80% ($v/v$) methanol and incubated in a water bath at 70 °C for 30 min followed by the addition of an equal volume of 5% phenol and 1.5 mL of concentrated sulfuric acid and incubation in the dark for 30 min. The absorbance of the reaction mixture was measured at 490 nm and the concentration of soluble sugars in the samples was calculated from the standard curve of glucose prepared with 100–1000 μg/mL concentration. The protein and amino acid content from each extract was estimated according to the method of Lowry et al. [42] and Yemm et al. [43], respectively.

### 2.6.6. Analysis of Plant Nutrients and Mineral Content

Extract of seedlings of plants from each treatment pots was used for the estimation of plant nutrient contents. Available nitrogen was estimated according to the method of Kjeldahl [24], P content was estimated according to the method of Holliday and Gartner [25] and K content was measured according to the method of Baghel [27]. The concentrations of Ca, Mg, K, Fe, Zn, Cu, Mn, and B were estimated according to the method of Gupta and Mohopatra [28].

All the experiments were carried out in five replicates. All the statistical analyses for wheat and maize plant parameters were performed separately. The average values of five replicates of wheat and maize were analyzed by one-way analysis of variance (ANOVA) followed by Tukey's test [44] through Statistix 8.1 software.

## 3. Results

*3.1. Screening for Salinity Stress Tolerance*

The effect of varying levels of various salts revealed that *K. variicola* SURYA grew well over the range of various salts. It grew luxuriously at a higher concentration of various salts such as NaCl (160 mM), KCl (80 mM), $Na_2CO_3$ (80 mM), and $NaHCO_3$ (100 mM) (Figure 1). However, this organism tolerated higher concentrations of NaCl (160 mM) compared to the other salts. The threshold level of NaCl that inhibited the growth of the organism was higher (>160 mM) as compared to the lowest threshold level (100 mM) of other salts.

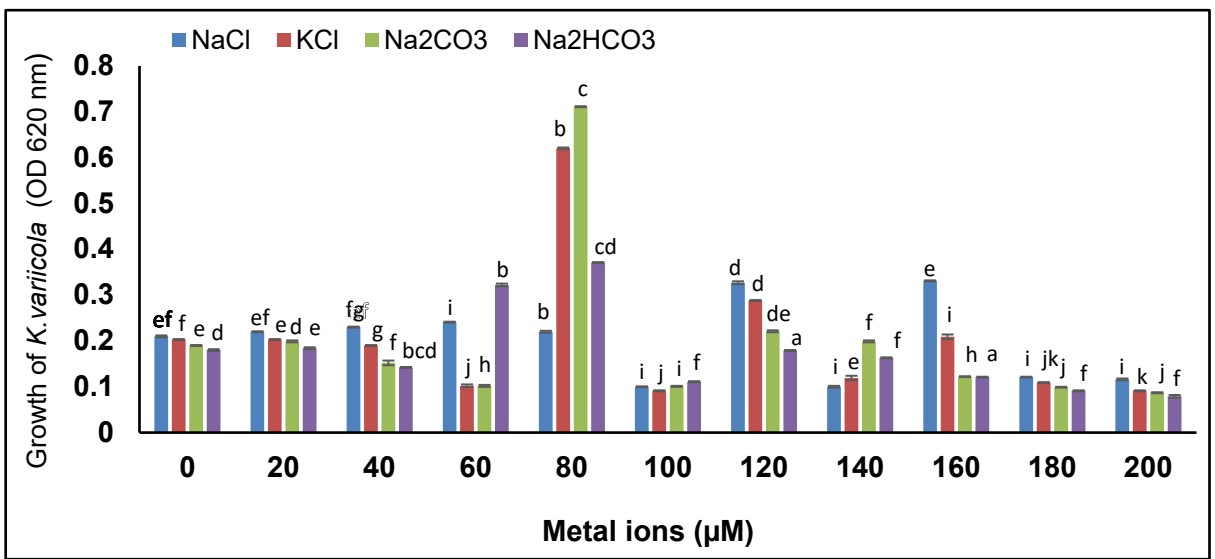

**Figure 1.** Effect of salinity stress on the growth of *K. variicola* SURYA6 after 24 h growth in NB amended with various salts (0–200 mM). Growth (OD at 620 nm) was measured spectrophotometrically with control. Values are the mean of five replicates and were analyzed by one-way ANOVA followed by Tukey's HSD test. Different letters indicate significant differences in the values at $p < 0.05$.

*3.2. Screening and Production of Salinity Ameliorating Traits*

*K. variicola* SURYA6 produced three principal salinity ameliorating traits namely ACCD, IAA, and EPS. It exhibited the ACCD activity of $6.01 \pm 1.9$ M/mg/h and produced $78.45 \pm 1.9$ µg/mL IAA, and $32.2 \pm 1.2$ g/L EPS. The viscosity of the inoculated medium increased from 10 to 97 cp while the viscosity of the control flask (un-inoculated) remained unchanged (10 cp).

*3.3. Confirmation of Non-Pathogenicity of K. variicola SURYA6*

Growth of *K. variicola* SURYA6 on MH medium revealed its sensitivity to the majority of the antibiotics tested; it exhibited sensitivity towards 26 different types of antibiotics and a small degree of resistivity towards only a few antibiotics such as cefuroxime, linezolid, ceftazidime, lincomycin, cloxacillin, and nitrofurantoin (Table 1). On blood agar, no growth of *K. variicola* SURYA6 and no hemolysis zone were observed. No release of oxygen effervescences from $H_2O_2$ solution after adding the *K. variicola* SURYA6 colony and no formation of intense deep-purple hue on TMPD impregnated filter paper was observed.

**Table 1.** Antibiotic sensitivity profile of *K. variicola* SURYA6

| Antibiotics | Symbol | Diameter Zone (mm) | Sensitivity/ Resistivity | Diameter Zone (mm) | Antibiotics | Symbol | Diameter Zone(mm) | Sensitivity/ Resistivity | Diameter Zone(mm) |
|---|---|---|---|---|---|---|---|---|---|
| Amikacin | AK | 14–17 | S | 20 | Co-Trimoxazole | COT | 10–16 | S | 25 |
| Amoxyclav | AMC | 13–18 | INT | 18 | Doxycycline HCL | DO | 12–16 | INT | 16 |
| Ampicillin | A/S | 11–15 | S | 22 | Ertapenem | Etp | 18–22 | INT | 20 |
| Azithromycin | AZM | 13–18 | INT | 18 | Gentamicin | GEN | 12–15 | S | 21 |
| Carbenicillin | CB | 19–23 | INT | 23 | Imepenem | IPM | 13–16 | S | 24 |
| Cefaperazone | SCF | 14–20 | S | 32 | Levofloxacin | LE | 13–17 | INT | 14 |
| Cefazolin | CZ | 14–18 | S | 27 | Linezolid | LZ | 20–21 | INT | 21 |
| Cefepime | CPM | 14–18 | S | 25 | Meropenem | MEM | 13–16 | S | 24 |
| Cefpirome | CFP | 19–23 | S | 28 | Norfloxacin | NOR | 12–17 | S | 20 |
| Ceftazidime | CAZ | 14–18 | S | 28 | Novobiocin | NV | 17–12 | INT | 16 |
| Ceftazidime | CTZ | 14–18 | S | 28 | Ofloxacin | OFX | 12–16 | S | 22 |
| Ceftizoxime | CZX | 14–20 | S | 24 | Piperacillin/ Tazo. | PIT | 17–21 | S | 28 |
| Ceftriaxone | CFS | 13–18 | S | 32 | Polymyxin B | PB | 8–12 | INT | 12 |
| Cefuroxime | CXM | 14–18 | R | 10 | Teicoplanin | TEI | 10–14 | R | 10 |
| Cephadroxil | CFR | 12–18 | R | 11 | Tetracycline | TE | 14–19 | S | 22 |
| Chloramphenicol | C | 13–17 | S | 19 | Ticarcillin | TI | 14–20 | R | 10 |
| Ciprofloxacin | CIP | 15–21 | S | 36 | Tigecycline | TGC | 7–13 | S | 22 |
| Clindamycin | CD | 14–21 | R | 14 | Tobramycin | TOB | 12–15 | INT | 15 |
| Colistin | CL | 10–11 | S | 16 | Ceftriaxone | CTR | 18–22 | S | 32 |
| Nalidixic Acid | NA | 14–18 | S | 26 | Cefixime | CFM | 16–18 | S | 32 |
| Nitrofurointoin | NIT | 12–14 | R | 11 | Ceftazidime/ clavu | CAC | 18–20 | S | 24 |

S = sensitive, INT = intermediate, R = resistant. *K. variicola* SURYA6 was grown on MH media having antibiotic discs on the surface. Plates were incubated at 28 ± 2 °C for 24–48 h and observed for the presence or absence of zones of growth inhibition. Values of sensitivity or resistivity of the organism to various antibiotics are the mean of five replicates and were analyzed by one-way ANOVA followed by Tukey's HSD test.

### 3.4. Plant Growth Promotion Studies—Pot Assay

Salinity affected the seed germination, root and shoot growth, and chlorophyll content in non-bacterized seeds of wheat and maize. Seed germination in wheat and maize decreased by 17.1% and 10% respectively compared to the non-bacterized seeds sown under normal soil (Figure 2). Root length, shoot height, and chlorophyll content in wheat seedling after 45 DAS was found reduced in the following order 32.1%, 19.7%, and 31.1% respectively. The application of *K. variicola* SURYA6 significantly improved these growth parameters. A 33.9% and 36% more seed germination in wheat and maize was evident over non-bacterized seeds sown in normal soil. Bacterized seeds sown under salinity showed significant improvement in root length, shoot height (Figure 3), and chlorophyll content (Figure 4) in wheat and maize seedling after 30 DAS and 45 DAS was evident due to bacterization as compared to the other treatments. A 111%, 35.1%, 64.8%, and 78.9%, and 85.1%, 99.0%, and 68.4% rise in root length, shoot height, and chlorophyll content was evident under salinity stress in the bacterized wheat and maize seeds respectively over the non-bacterized seeds under normal soil at 45 DAS. Application of *K. variicola* SURYA6 improved the seed germination and plant growth parameters under normal soil as well as saline soil, after 30 DAS as well as 45 DAS however, the improvement in seed germination and plant growth was higher under salt stress conditions as compared to normal soil conditions after 30 DAS as well as 45 DAS.

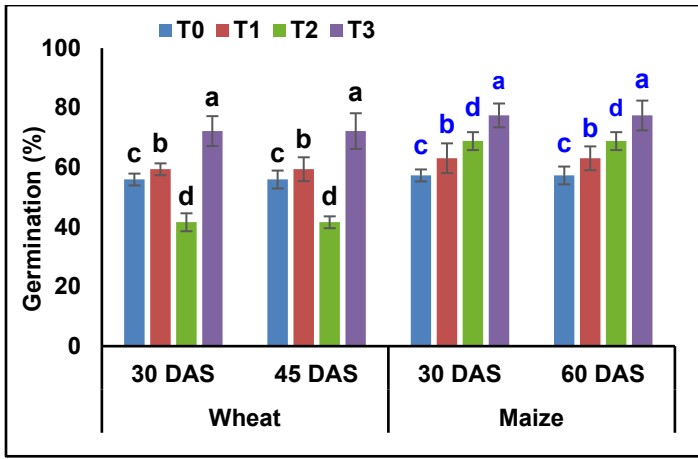

**Figure 2.** Effect of *K. variicola* SURYA6 inoculation on seed germination in wheat and maize. Values are the mean of five replicates and were analyzed by one-way ANOVA followed by Tukey's HSD test. The values were initially transformed into arcsine values. Different letters within each plant species indicate significant differences in the values at $p < 0.05$. T0 = Non-bacterized seeds sown in normal soil; T1 = Bacterized seeds sown in normal soil. T2 = Non-bacterized seeds sown in saline soil; T3 = Bacterized seeds sown in saline soil.

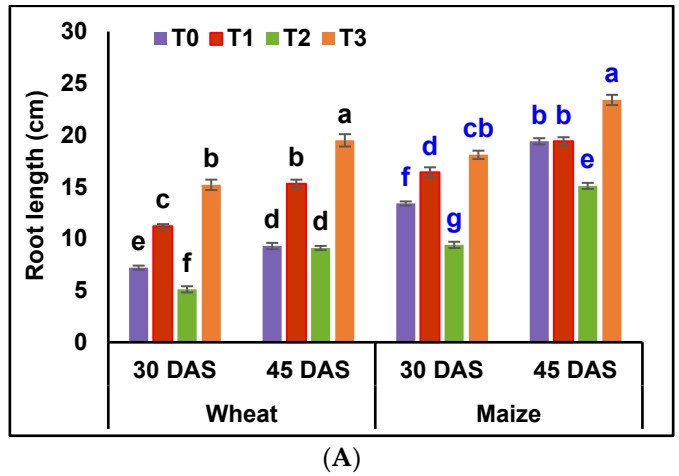

(A)

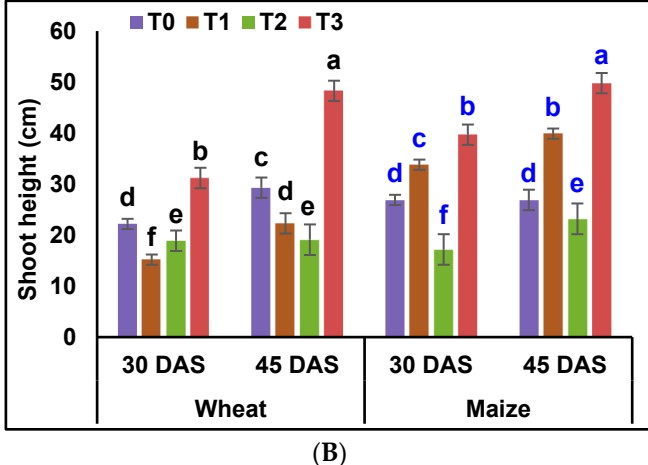

(B)

**Figure 3.** (**A**) Effect of *K. variicola* SURYA6 inoculation on root length in wheat and maize at 30, and 45 DAS. (**B**) Effect of *K. variicola* SURYA6 inoculation on shoot height in wheat and maize at 30, and 45 DAS. Different letters within each plant species indicate significant differences in the values at $p < 0.05$.

*3.5. Analysis of Osmolytes and Biochemical Contents in Wheat and Maize Plants*

Saline soil caused a substantial decrease in osmolyte, proteins, and amino acid contents in non-bacterized wheat and maize plants. After 45 DAS of non-bacterized wheat and maize seeds in saline soil, the proline, sugar, protein, and amino acids content were found reduced in the following order 21.2%, 64.0%, 39.2%, and 45.8%, and 45.5%, 39.1%, 10.4%, and 17.2% respectively. While the application of *K. variicola* SURYA6 resulted in a significant improvement in osmolytes, proteins, and amino acid contents in wheat and maize grown under saline soil as well as normal soil. The application of *K. variicola* SURYA6 significantly improved the osmolyte and biochemical contents in plants grown under salinity (Figure 5). A 50.0% (Figure 5A), 37.1% (Figure 5B), 52.0% (Figure 5C), and 309.9% (Figure 5D), 245.5% (Figure 5A), 78.3% (Figure 5B), 86.0% (Figure 5C), and 146.4% (Figure 5D) increase proline, sugars, protein, and amino acid contents in wheat and maize seedlings at 45 DAS was

evident over the control. The improvement in the level of these biomolecules was higher under salinity as compared to normal soil conditions.

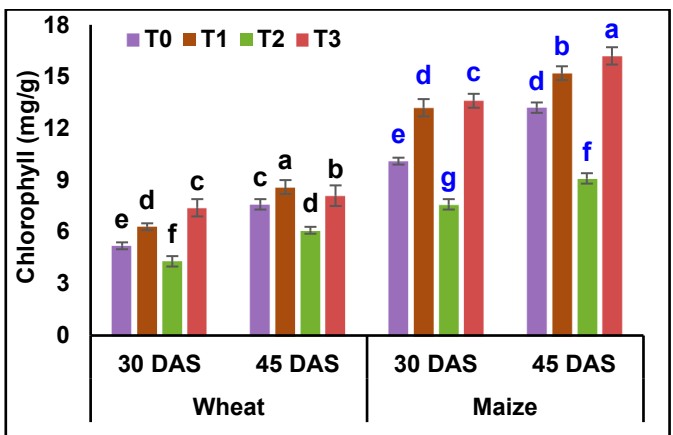

**Figure 4.** Effect of *K. variicola* SURYA6 inoculation on chlorophyll content in wheat and maize at 30 and 45 DAS. Values are the mean of five replicates and were analyzed by one-way ANOVA followed by Tukey's HSD test. The values were initially transformed into arcsine values. Different letters within each plant species indicate significant differences in the values at $p < 0.05$. T0 = Non-bacterized seeds sown in normal soil; T1 = Bacterized seeds sown in normal soil. T2 = Non-bacterized seeds sown in saline soil; T3 = Bacterized seeds sown in saline soil.

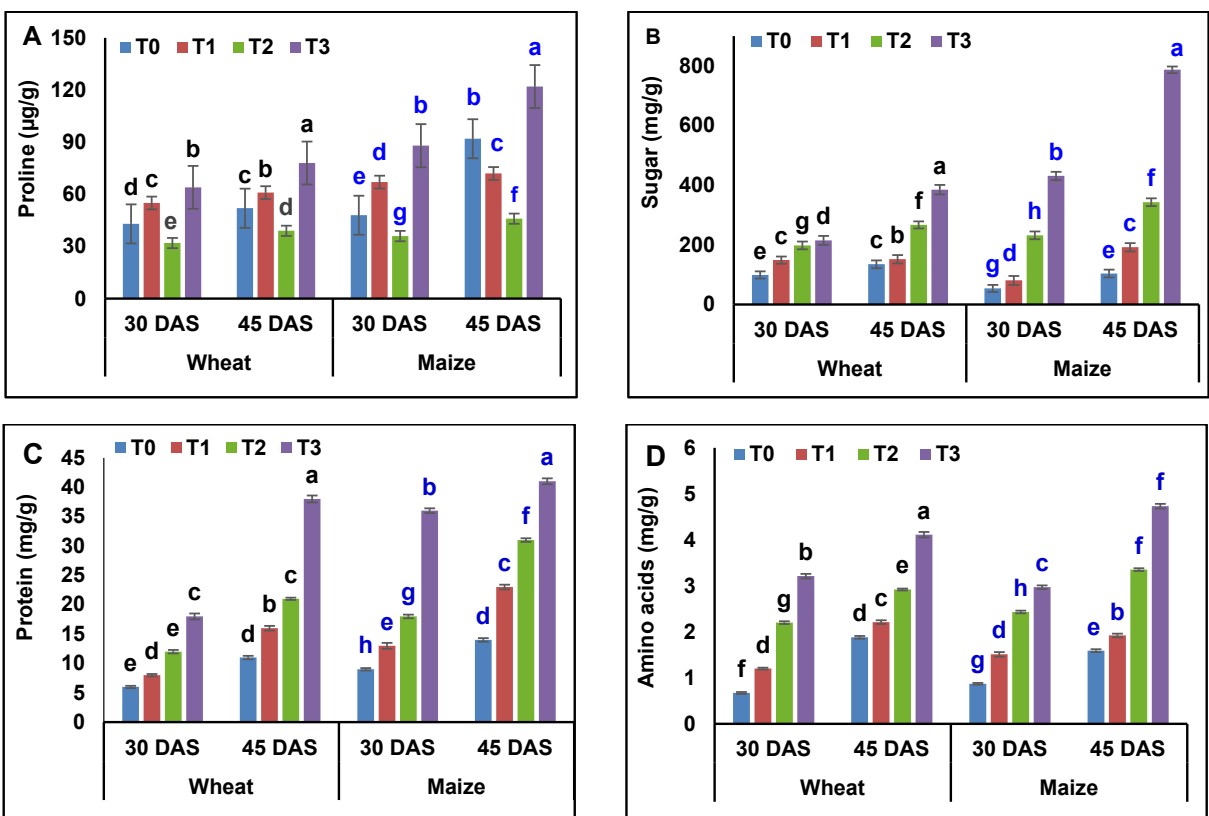

**Figure 5.** Effect of *K. variicola* SURYA6 inoculation on (**A**) proline, (**B**) sugar, (**C**) protein, and (**D**) amino acid contents at 30 DAS and 45 DAS in wheat and maize. Mean values of five replicate were analyzed by one-way ANOVA followed by Tukey's HSD test. Different letters within each plant species indicate significant differences in the values at $p < 0.05$. T0 = Non-bacterized seeds sown in normal soil; T1 = Bacterized seeds sown in normal; T2 = Non-bacterized seeds sown in saline soil; T3 = Bacterized seeds sown in saline.

*3.6. Analysis of Mineral Content of Wheat and Maize Plants*

The application of *K. variicola* SURYA6 caused significant improvement in N, P, Na, K, and Mg contents in wheat and maize under salinity. It resulted in 35.1% to 89.9% increase in these mineral ions vis a-vis uninocluated plants under salinity (Table 2). The improvement in the level of these minerals was higher under salinity conditions as compared to normal soil.

**Table 2.** Effect of *K. variicola* SURYA6 inoculation on mineral contents in wheat and maize.

| Mineral Content | Seedling | DAS | Before Sowing | After Sowing | | | |
|---|---|---|---|---|---|---|---|
| | | | | T0 | T1 | T2 | T3 |
| Total Nitrogen (%) | Wheat | 30 | 2.01 ± 0.02a | 1.35 ± 0.03d | 2.99 ± 0.06a | 2.31 ± 0.02a | 3.91 ± 0.08e |
| | | 45 | 3.03 ± 0.05b | 2.19 ± 0.0e | 1.67 ± 0.02c | 3.78 ± 0.05d | 4.99 ± 0.07b |
| | Maize | 30 | 2.81 ± 0.03c | 1.82 ± 0.06b | 3.15 ± 0.07d | 3.91 ± 0.03c | 4.01 ± 0.09d |
| | | 45 | 3.94 ± 0.04c | 2.67 ± 0.07d | 4.27 ± 0.02d | 4.87 ± 0.04c | 6.22 ± 0.04a |
| P (%) | Wheat | 30 | 1.9 ± 0.04b | 0.99 ± 0.05b | 2.01 ± 0.02d | 2.8 ± 0.04b | 2.61 ± 0.06a |
| | | 45 | 2.5 ± 0.03d | 1.52 ± 0.04d | 3.12 ± 0.07a | 3.4 ± 0.03d | 3.81 ± 0.03b |
| | Maize | 30 | 1.87 ± 0.05d | 1.05 ± 0.03b | 2.91 ± 0.06c | 2.79 ± 0.05d | 3.41 ± 0.02d |
| | | 45 | 2.78 ± 0.06d | 1.99 ± 0.08c | 3.66 ± 0.04d | 4.82 ± 0.06d | 4.89 ± 0.04a |
| Na (%) | Wheat | 30 | 1.12 ± 0.02d | 0.85 ± 0.06d | 1.99 ± 0.04c | 2.92 ± 0.02c | 2.85 ± 0.11d |
| | | 45 | 2.08 ± 0.03b | 1.85 ± 0.03d | 2.88 ± 0.02d | 4.48 ± 0.03b | 3.89 ± 0.13a |
| | Maize | 30 | 2.01 ± 0.05c | 1.42 ± 0.02b | 2.99 ± 0.06d | 3.11 ± 0.05c | 3.32 ± 0.15c |
| | | 45 | 2.31 ± 0.06c | 1.32 ± 0.05c | 3.01 ± 0.12a | 4.42 ± 0.06b | 3.12 ± 0.09a |
| K (%) | Wheat | 30 | 1.32 ± 0.04b | 0.99 ± 0.03c | 2.01 ± 0.07c | 2.41 ± 0.04b | 3.21 ± 0.07c |
| | | 45 | 2.27 ± 0.01b | 1.97 ± 0.02d | 3.32 ± 0.03b | 4.87 ± 0.01b | 4.31 ± 0.05a |
| | Maize | 30 | 2.31 ± 0.06b | 1.42 ± 0.06b | 2.99 ± 0.08a | 3.42 ± 0.06b | 3.32 ± 0.04b |
| | | 45 | 2.31 ± 0.05b | 1.42 ± 0.04b | 2.99 ± 0.02a | 4.54 ± 0.05b | 3.32 ± 0.08b |
| Mg (%) | Wheat | 30 | 0.51 ± 0.05a | 0.11 ± 0.02c | 1.01 ± 0.07b | 0.91 ± 0.05a | 1.51 ± 0.06e |
| | | 45 | 0.71 ± 0.08b | 0.11 ± 0.06c | 0.75 ± 0.04d | 1.02 ± 0.08b | 1.99 ± 0.02e |
| | Maize | 30 | 0.97 ± 0.03d | 0.48 ± 0.03b | 1.75 ± 0.06a | 1.27 ± 0.03d | 1.99 ± 0.05e |
| | | 45 | 1.75 ± 0.07d | 0.98 ± 0.05d | 2.61 ± 0.08c | 4.94 ± 0.07a | 2.87 ± 0.04a |

Values are the mean of five replicates analyzed by one-way ANOVA followed by Tukey's HSD test. Different letters within each plant species indicate significant differences in the values at $p < 0.05$. T0 = Non-bacterized seeds sown in normal soil; T1 = Bacterized seeds sown in normal; T2 = Non-bacterized seeds sown in saline soil; T3 = Bacterized seeds sown in saline.

*3.7. Analysis of Soil Physical Parameters and Nutrients*

Inoculation of *K. variicola* SURYA6 under salinity improved the physico-chemical parameters of soil as compared to the un-inoculated soil (control) and bacterization in normal soil. This inoculation reduced the electrical conductivity of soil from 5.7 to 4.0 and pH from alkalinity (9.2) to neutrality (6.8) and significantly improved the soil nutrients. Soil organic carbon, nitrogen, P, K, $Ca^{2+}$, $Mg^{2+}$, S, $Fe^{2+}$, $Mn^{2+}$, $Zn^{2+}$, Cu, and B increased by 312.90%, 193.63%, 180.42%, 177.08%, 220.60%, 188.50%, 150.80%, 157.65%, 129.44%, 122.17%, 142.65%, and 131.25%, respectively (Table 3). Seed bacterization improved the soil nutrient contents of normal as well as saline soil. However, the improvement in nutrient contents was higher in saline soil that received the inoculum as compared to the saline soil that did not receive the inoculum and normal soil that received the inoculum.

**Table 3.** Analysis of physicochemical parameters of the soil before sowing and at 45 DAS

| | Before Sowing | After Sowing | | | |
|---|---|---|---|---|---|
| | | T0 | T1 | T2 | T3 |
| Electrical conductivity (dS/m) | 4.0±0.10a | 3.9± 0.03e | 4.2 ± 0.09d | 4.7 ± 0.07c | 5.7 ± 0.11b |
| pH | 9.2± 0.11a | 7.7± 0.11g | 7.3 ± 0.09d | 6.9 ± 0.10f | 6.8 ± 0.12o |
| Organic carbon (%) | 0.93± 0.12c | 0.93 ± 0.12c | 1.51 ± 0.13b | 0.43 ± 0.11e | 2.91 ± 0.14c |
| Available nitrogen (Kg/ha) | 50.87 ± 0.13d | 50.87 ± 0.13d | 68.57 ± 0.12d | 25.87 ± 0.11b | 98.57 ± 0.10c |
| Available P (Kg/ha) | 26.16 ± 0.14d | 26.16 ± 0.14a | 27.21 ± 0.11c | 17.16 ± 0.12a | 47.21 ± 0.13b |
| Available K (Kg/ha) | 44.11 ± 0.13a | 44.11 ± 0.13c | 56.12 ± 0.10d | 34.22 ± 0.13c | 78.12 ± 0.12b |
| Available $Ca^{++}$ (ppm) | 1650 ± 0.09d | 1650 ± 0.09c | 2640 ± 0.09d | 1230 ± 0.12a | 6400 ± 0.13c |
| Available $Mg^{++}$ (ppm) | 400 ± 0.05d | 400 ± 0.05c | 900 ± 0.08d | 300 ± 0.10b | 1254 ± 0.05c |
| Available $S^{++}$ (ppm) | 21.67 ± 0.04c | 21.67 ± 0.04d | 39.68 ± 0.07a | 15.67 ± 0.11d | 51.68 ± 0.10a |
| Available $Fe^{++}$ (ppm) | 11.1 ± 0.02c | 11.1 ± 0.02c | 15.5 ± 0.06b | 8.1 ± 0.12 a | 17.5 ± 0.09c |
| Available $Mn^{++}$ (ppm) | 19.7 ± 0.03a | 19.7 ± 0.03c | 12.5 ± 0.08c | 7.7 ± 0.13d | 25.5 ± 0.07b |
| Available $Zn^{++}$ (ppm) | 6.81 ± 0.09a | 6.81 ± 0.09d | 8.32 ± 0.07c | 5.81 ± 0.10c | 8.32 ± 0.06d |
| Available $Cu^{++}$ (ppm) | 4.45 ± 0.11b | 4.45 ± 0.11b | 6.85 ± 0.09c | 4.45 ± 0.13a | 7.22 ± 0.10d |
| Available B (ppm) | 0.05 ± 0.09a | 0.05 ± 0.09c | 0.28 ± 0.06c | 0.03 ± 0.11c | 1.35 ± 0.12b |

Values are the mean of five replicates analyzed by one-way ANOVA followed by Tukey's test. Values are the mean of five replicates analyzed by one-way ANOVA followed by Tukey's HSD test. Different letters indicate significant differences in the values at $p < 0.05$. T0= non-bacterized seeds sown in normal soil. T1 = Bacterized seeds sown in normal soil. T2 = non-bacterized seeds sown in saline soil. T3 = bacterized seeds sown in saline soil.

## 4. Discussion

### 4.1. Screening for Salinity Stress Tolerance

Halophilic microorganisms possess the ability to utilize a wide range of salts. The ability of *K. variicola* SURYA6 to grow at high concentrations of various salts can be attributed to the excretion of EPS that maintains the viability of bacterial cells under salt stress [33,43,44]. This indicates its halophilic physiology and the ability of *K. variicola* SURYA6 to survive under salinity stress. Halophilic organisms grow at higher salt levels as they can balance the osmotic pressure of the cytoplasm according to the environment and thus protect their enzymes and proteins [45]. Sharma et al. [6] reported the tolerance to higher levels (4 to 8% NaCl) of salts in halophilic *Klebsiella* sp. isolated from the plant roots. Singh et al. [46] reported the growth of *Klebsiella* sp. at a higher level (6%) of salt (NaCl). Inhibition of growth of *K. variicola* SURYA6 above threshold level i.e., 70 mM of NaCl and 30 mM of other salts is because of the denaturation of the proteins, enzymes, and damages to the cell membrane of the organism.

### 4.2. Screening and Production of Salinity Ameliorating Traits

PGPR that colonizes plant protect host plants during various abiotic stresses [47]. Stress tolerance is enhanced in plants by the PGPR through the production of IAA, ACCD, and EPS and the improved nutrient content, thus improving the health of plants under salt stress [48,49]. Kruasuwan and Thamchaipen [50] observed the ACCD activity of 0.1 μM/mg/h in endophytic *Enterobacter* sp. EN-21 and reported this PGPR as an effective inoculant for salinity stress management. Singh et al. [47] reported ACCD activity in *Klebsiella* sp. SBP-8, isolated from the sorghum rhizosphere in the desert region of Rajasthan, India. They observed optimum growth and ACCD activity at high (6%) concentrations of salt, indicating its potential to survive and associate with plants under saline soil. Kim et al. [51] reported the production of copious amounts of IAA in *K. variicola* AY-13. Mondal et al. [52] reported the production of EPS in *Klebsiella variicola* ATCC BAA-830. Sagar et al. [16] reported the production of various PGP traits and ACCD in *E.cloacae* PR4.

### 4.3. Confirmation of Non-Pathogenicity of K.variicola SURYA6

Multiple resistances to a wide variety of antibiotics are commonly exhibited by *Klebsiella* sp. associated with clinical infections [53]. However, the isolate, *K. variicola* SURYA6 exhibited sensitivity towards 26 different types of antibiotics. Moreover, the inability of the isolate to grow on BA confirmed its non-clinical and non-pathogenic origin. Some

reports showed the *K.variicola* is associated with extended-spectrum beta-lactamase (ESBL) producing ability but *K. variicola* SURYA6 did not show ESBL activity [54]. Barrios et al. [23] claimed *K. variicola* as a versatile bacterium capable of colonizing different hosts such as plants, humans, insects, and animals. *K.variicola* of plant origin is non-pathogenic to humans or animals [53,54].

### 4.4. Plant Growth Promotion Studies—Pot Assay

Seed germination and plant growth are adversely affected due to the presence of even low (1 M) stress of salts [15,16]. Sachdev et al. [47] reported the improved germination in wheat variety lokwan due to inoculation of IAA producing *K. pneumonae* [47]. They reported 92.71% increase in seed germination, 1.59-fold increase in root length and 1.52-fold increase in shoot length in bacterized wheat seedlings as compared to the control. Singh et al. [47] reported improved seed germination and plant growth in mitigation of salinity in wheat inoculated with *Klebsiella* sp. SBP-8 *K. variicola* SURYA6 bacterized seeds of wheat and maize promoted seed germination and overall growth of plants under high stresses of a wide variety of salts. This is because halophilic PGPR is known to reduce the salt level and protect the plant from osmotic effects, thus helping the growth of the plant under salt stress conditions while ameliorating soil salinity [16]. The ability of the *K. variicola* SURYA6 to promote seed germination and plant growth in wheat and maize under salinity stress reflects its bio-efficacy as an effective bioinoculant for wheat and maize under saline soil. Sagar et al. [8,15] reported the negative effect of NaCl on seed germination but the application of ACCD positive halophilic *Enterobacter* sp. PR14 promoted 50–90% more seed germination under salt stress (30 mM NaCl). Excretion of ACCD by PGPR under slat stress is one of the principal mechanisms of salt tolerance [16]. The enzyme lowers the level of ACC that is exuded by plant roots.

### 4.5. Analysis of Osmolytes and Biochemical Contents in Wheat and Maize Plants

Singh et al. [55] observed that inoculation of wheat with *Klebsiella* sp. SBP-8 under salt stress results in increased root length, shoot height, and fresh and dry weight of seedling. The increased rate of seed germination in *K. variicola* SURYA6 treated seeds from test pot may be due to the release of growth hormone IAA [47]. There are reports which support the positive effect of multifarious PGPR on the growth promotion in various crop plants [56]. *Klebsiella* sp. showed 1.5-fold yield enhancement in maize during field trials [57,58].

### 4.6. Analysis of Mineral Content of Wheat and Maize Plants

Improvement in plant growth parameters such as roots, shoots, and grains of wheat plants and increase in soil fertility following the inoculation with multifarious PGPR is a well-known feature of many PGPR including *Klebsiella* sp. [34]. The increase in chlorophyll content in wheat and maize is because of the increase in protein, amino acid, and Mg contents in plants and an increase in soil nutrient contents. The increase in plant protein, amino acids, and Mg level in the soil contributes to more synthesis of chlorophyll. The increase in chlorophyll content and photosynthetic activity in response to PGPR has been reported by Kaur and Reddy [59]. Production of phytohormone especially IAA in *Azospirillum brasilense* has been reported to relieve the damaging effects of NaCl [60]. Albacete et al. [61] observed that plants inoculated with IAA producing PGPR exhibited higher root and leaf growth which is considered as an adaptive response to salt stress. IAA producing PGPR strains were also reported to enhance nutrient uptake under hydroponic conditions [62]. Singh et al. [47] reported the amelioration of salt stress in wheat and improvement in plant growth and chlorophyll content in wheat inoculated with salt-tolerant *Klebsiella* sp. SBP-8. They found that ACCD positive *Klebsiella* sp. SBP-8 protects the plant from salt stress by removing excess $Na^+$ and absorbing maximum $K^+$. Kim et al. [51] found that the inoculation of *K. variicola* AY significantly improved the growth and chlorophyll contents in soybean. Secretion of EPS has been reported to reduce toxic ion accumulation and enhancement of $K^+/Na^+$ ratio in plants [63]. Ashraf et al. [64] reported a decrease in $Na^+$

accumulation in wheat and maize plants due to the excretion of EPS produced by that *Aeromonas hydrophilacaviae* and *Bacillus* sp. The EPS was found to bind $Na^+$ in roots and prevent their transfer to leaves. Similarly, EPS producing *B. circulans* and *B. polymyxa* were found to enhance $K^+/Na^+$ and $Ca^{2+}/Na^+$ ratio during salt stress conditions. This effect was the result of the cation chelating property of EPS [65]. Upadhyay et al. [66] reported wheat rhizosphere PGPR to produce EPS that significantly decreased $Na^+$ uptake in plants under normal as well as saline conditions. Kumari et al. [67] claimed that EPS producing bacterial strains enhance $K^+/Na^+$ ratio which protects chloroplast and chlorophyll in soybean plants from osmotic damage under salt stress. PGPR that produces EPS plays a crucial role in the growth of plants during salt stress by forming hydrophilic biofilms [2]. EPS producing PGPR has the potential to ameliorate salt stress by making rhizosheaths around the plant root that chelate $Na^+$ ions. Adsorption of $Na^+$ by EPS reduces the toxicity of $Na^+$ making it unavailable for plants. *Bacillus subtilis* inoculation to *Arabidopsis thaliana* reduces the influx of $Na^+$ through downregulating the expression of $HKT1/K^+$ transporter [68]. The results of the present studies are in line with the results obtained by Arora et al. [6] they reported EPS produced by PGPR under salt stress binds Na ions and reduces its toxic effects. Nia et al. [10] reported a higher accumulation of proline and $Na^+$ and protein contents in leaves and shoots respectively in plants inoculated with halophilic *Azospirillum* as compared to the controls and plants inoculated with non-halophilic *Azospirillum*. Under salinity stress, plants accumulate proline and soluble sugars and ions to maintain osmotic adjustment [69]. They claimed the better growth of *Azospirillum* inoculated wheat under salinity stress is the result of increased levels of proline, nitrogen, and protein in the leaves of inoculated plants. PGPR like *Azospirillum* accumulate more proline and glutamic acid in response to salt stress in wheat and maize, thus acting as an osmoprotectant. PGPR help to restore and improve intracellular nitrogen level which is declined due to salinity stress [70]. Nia et al. [10] found that improved growth of wheat plants under salt stress was due to an increase in photosynthetic pigments and more accumulation of solutes.

### 4.7. Analysis of Soil Physical Parameters and Nutrients

The increase in soil nutrient content due to inoculation with *Klebsiella* sp. has been reported by Balaban et al. [70]. They found that the increase in the concentrations of 18 different soil microelements was due to inoculation with *Klebsiella* sp. Nadeem et al. [71] Claimed the significant changes in C, Zn, Fe, Mn, and Cu content of soil inoculated with *Klebsiella* sp. This indicates the role of *Klebsiella* sp. in improving soil fertility. An increase in macro and micronutrient contents in the soil that received *K. variicola* SURYA6 inoculation also supports the theory of an increase in soil fertility due to the inoculation of PGPR [53].

### 5. Conclusions

The improvement in plant growth parameters under salt stress conditions is attributed to the ability of PGPR to excrete salinity ameliorating metabolites such as ACCD, IAA, EPS, and osmoprotectants that help in plant growth and relief from the damaging effects of salt stress. Multifarious halotolerant *K. variicola* SURYA6 can function as effective bioinoculants for plant growth promotion, protect plants from osmotic damages caused due to excess salt, help in ameliorating soil salinity, and enrich the soil nutrients. However, repeated field trials are required to confirm the bio-efficacy potential of *K. variicola* SURYA6.

**Author Contributions:** S.P.K.: Methodology; Writing-Original draft; Y.C.A.: Project administration and supervision; R.Z.S.: Writing—review the original draft and review and editing; H.E.E., Review, and editing of the manuscript; S.Z.H., Formal analysis; N.I., Review, and editing of the manuscript, statistical analysis, and preparation of graphs; A.M.E.: Formal analysis; and N.M. and A.H.B.: funding acquisition. All authors have read and agreed to the published version of the manuscript.

**Funding:** The Researchers Supporting Project number (RSP-2020/15), King Saud University, Riyadh, Saudi Arabia and Universiti Teknologi Malaysia (UTM) through project (nos. QJ130000.3609.02M43, QJ130000.3609.02M39) and Allcosmos Industries Sdn. Bhd. Through research project (no. R.J130000.7344.4B200).

**Institutional Review Board Statement:** Not Applicable.

**Informed Consent Statement:** Not Applicable.

**Data Availability Statement:** All the data is available in the paper.

**Acknowledgments:** The authors extend their appreciation to The Researchers Supporting Project number (RSP-2020/15), King Saud University, Riyadh, Saudi Arabia, and Universiti Teknologi Malaysia (UTM) for project (nos. QJ130000.3609.02M43 and QJ130000.3609.02M39), and All Cosmos Industries Sdn. Bhd. (project no. R.J130000.7344.4B200) for funding this work.

**Conflicts of Interest:** The authors declare no conflict of interest.

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
