# Peer review of "Inoculation of Klebsiella variicola Alleviated Salt Stress and Improved Growth and Nutrients in Wheat and Maize"

_agronomy, doi:10.3390/agronomy11050927_

Round 1
Reviewer 1 Report
Most of the comments/suggestions are included in the reviewed pdf document (attached). In addition, a major revision in the statistical data analysis is required. The authors mentioned that the pot assay was done in a randomized block design (RBD) / randomized complete block design (RCBD) with 4 treatments and 5 replications. Therefore, the analysis of variance should also be based on RBD. However, the statistical analyses are not correct as they are not based on RBD as can be seen in the results section including figures and tables. Therefore, results, discussion and conclusion can change after the correct ANOVA model implementation. Please use replication as the block in the ANOVA model (Each dependent variable as a function of treatment and replication plus interaction if applicable) and use the data as such without averaging for 5 replications (for ANOVA). Please do not hesitate to get help from an expert statistician. In Table 2, so many letters can be seen to indicate significant differences, but that is not the correct way of showing significant differences. The correct way to do it for Table 2 is analyze for each row separately. For example, consider for Total N in wheat for 30 DAS row, perform ANOVA. Now, check the P-values for treatment and replication. If P-value for the treatment is significant, a Tukey's HSD test is required in addition. Now based on Tukey's HSD, at most four letters (a, b, c, and d) are enough to show significant differences in treatments (T0, T1, T2, and T3). Each next row can again use the letters a, b, c, and d as the comparison is among treatments within a row. Please also prepare an ANOVA table including P-values and HSD values where applicable. The manuscript has very good information and useful contents. A major revision in statistical analyses is needed.

Reviewer 1 Report
The authors are grateful to the reviewer for such an excellent review of the MSS. These comments helped a lot in the improvement of the MSS.
- Most of the comments/suggestions are included in the reviewed pdf document (attached).
Authors’ response: The authors are agreed to all the comments and suggestions of the reviewer. All suggestions included in the reviewed pdf document have been incorporated in the MSS file. Corrections are done at Line No. 26,26,43,45,102,432,433….
- In addition, a major revision in the statistical data analysis is required. The authors mentioned that the pot assay was done in a randomized block design (RBD) / randomized complete block design (RCBD) with 4 treatments and 5 replications. Therefore, the analysis of variance should also be based on RBD. However, the statistical analyses are not correct as they are not based on RBD as can be seen in the results section including figures and tables. Therefore, results, discussion, and conclusion can change after the correct ANOVA model implementation. Please use replication as the block in the ANOVA model (Each dependent variable as a function of treatment and replication plus interaction if applicable) and use the data as such without averaging for 5 replications (for ANOVA). Please do not hesitate to get help from an expert statistician.
Authors’ response: The pot assay was done in a randomized block design (RBD) / randomized complete block design (RCBD) with 4 treatments and 5 replications. The data were statistically analyzed by analysis of variance based on RBD and followed by Turkey’s test. This information is already mentioned in the footnotes of all the figures and tables [Line No. Fig 2-5, Table 2 and 3]. The statistical analysis was performed by an expert statistician as mentioned in author's contribution.
- In Table 2, so many letters can be seen to indicate significant differences, but that is not the correct way of showing significant differences. The correct way to do it for Table 2 is to analyze each row separately. For example, consider for Total N in wheat for 30 DAS row, perform ANOVA. Now, check the P-values for treatment and replication. If P-value for the treatment is significant, a Tukey's HSD test is required in addition. Now based on Tukey's HSD, at most four letters (a, b, c, and d) are enough to show significant differences in treatments (T0, T1, T2, and T3). Each next row can again use the letters a, b, c, and d as the comparison is among treatments within a row.
Authors’ response: In Table 2 and Table 3 to indicate significant differences, only four letters (a, b, c, and d) are now mentioned in each row. Values are the mean of five replicates analyzed by one-way ANOVA followed by Turkey’s HSD test [Line 333, 352]
- The manuscript has very good information and useful contents.
Authors’ response: The authors are thankful to the Reviewer for appreciation.
- A major revision in statistical analyses is needed.
Authors’ response: Statistical analyses are thoroughly revised in all Figures and Tables

Reviewer 2 Report
This manuscript reports the effects of a PGPR train in alleviating salt stress in wheat and maize. The study is interesting and could be considered for publication after major revision.
Specific comments:
Title: consider to use the key finding as the title, such as something like “Inoculation of Klebsiella variicola Alleviated Salt Stress and Improved Plant Growth in Wheat and Maize”
Abstract:
need to rewrite - too long and not well organized. A good abstract may include 1-2 sentence(s) stating the background/aims of this study, 1-2 sentence(s) of the methodologies including bacteria/plant materials, key findings from these study, and one sentence of conclusion.
L95 “The present study aimed to…”
Introduction:
The first two paragraphs talking about general knowledge related to wheat and maize can be shortened.
Materials and Methods
The potting study in 2.1 and 2.2 does not show the experimental design including pot size, replications, treatments, growth environments, when to harvest, etc. All of these important information was missing.
L104. As with the K. variicola of choice for the study, it is essential to explain the rationale behind or criterion used for the selection of the two crop species, wheat and maize, and associated genotypes, so more data are expected.
L106-107 “It was earlier identified based on …” What does it mean? Not clear of this sentence.
L108: need a reference for the PGP traits
L109 “saline soil” from where, or how to add salt if modified?
L117 “contained leaf biomass” not clear?
Results
Show the key results supporting your main findings only. Some descriptions are not clears, such as L329-331, citing 10 percentages together for two crops and five elements is too complicated. You may talk about the increase trends and increment ranges.
L231-252 This information was missing from Methods section. Need to remove and add to Method.
From Fig.1 bar data show it even tolerated the highest concentration of all salt forms. Not much differences in the growth under 100 and 200 uM. This needs explanations in Discussion section.
Discussion
It would be helpful if the discussion is presented under several subtitles to highlight the main issues to be discussed.
L432. “These nutrients have been reported to reduce toxic ion accumulation and enhancement of K+/Na+ ratios in plants”. Which nutrients function to avoid toxin accumulation while balancing K+/Na+? How did you come to this from the above sentence? K+/Na+ should be “K+/Na+”, please check elsewhere.
Conclusion
First two sentences can be deleted – they are not conclusion from this study.
The writing needs lots of work, including grammatical errors and logical expressions. For example:
L23 not clear of what kind of “chemical approaches” often used.
L38 separate “seed germination” and “root length” with a coma
L39 It is better to rephrase “shoot length” as “shoot height”
L40 add coma both before and after the word “respectively”
L43-44 no need to use abbreviations when use only once.
L48 correct errors of “a higher concentrations”
L51 why use capital W for wheat? Also L96, L100. Please check throughout the manuscript.
L104-105 Grammatical error. Insert “and” before “were procured from…”
L117. Mixture that did not contain…,rather than “contained”
L436 “ratios” rather than “ratio”
L439 “K+/Na+” should be “K+/Na+ ratio” instead?
L439,452,454. Change “enhances”, ”accumulates”, and ”helps” into their plural form, respectively.
Reviewer 2 Report
This manuscript reports the effects of a PGPR train in alleviating salt stress in wheat and maize. The study is interesting and could be considered for publication after major revision.
Specific comments:
- Title: consider to use the key finding as the title, such as something like “Inoculation of Klebsiella variicola Alleviates Salt Stress and Improves Plant Growth in Wheat and Maize”
Authors’ response: Agreed. The title is revised as suggested
Abstract:
- need to rewrite - too long and not well organized. A good abstract may include 1-2 sentence(s) stating the background/aims of this study, 1-2 sentence(s) of the methodologies including bacteria/plant materials, key findings from these study, and one sentence of conclusion.
Authors’ response: Agreed. The abstract is now organized as per the suggestion and has been shortened from 464 words to 346 words
- L95 “The present study aimed to…”
Authors’ response: This sentence is revised as The present study was aimed to evaluate the plant growth promoting and ability of in wheat and maize under salinity stress, and its ability to enrich the soil nutrient contents under saline conditions. [Line no. 80]
Introduction
- The first two paragraphs talking about general knowledge related to wheat and maize can be shortened.
Authors’ response: The first two paragraphs about wheat and maize have been shortened.
Materials and Methods
- The potting study in 2.1 and 2.2 does not show the experimental design including pot size, replications, treatments, growth environments, when to harvest, etc. All of these important information was missing.
Authors’ response: All these details are already mentioned under heading No. 2.7.1. Experimental design. Harvesting period is now mentioned [Line No. 169-176]. Additional information is now provided [Line No. 181-182].
- As with the K. variicola of choice for the study, it is essential to explain the rationale behind or criterion used for the selection of the two crop species, wheat and maize, and associated genotypes, so more data are expected.
Authors’ response: Since K. variicola was obtained from wheat and maize rhizosphere and it produced multiple plant growth-promoting (PGP) traits based on the previous work of the author [Line No. 24] (Reference Kusale et al 2021 is cited in text and listed in Reference list at No. 24].
- L106-107 “It was earlier identified based on …” What does it mean? Not clear of this sentence.
Authors’ response: This sentence is deleted
- L108: need a reference for the PGP traits
Authors’ response: Reference No. [24] is added
- L109 “saline soil” from where, or how to add salt if modified?
Authors’ response: The text (prepared by adding 100 mM of NaCl per kg of soil) is added. The information of salin soil is already given under heading No, 2.7.2. Making of Saline Soil
- L117 “contained leaf biomass” not clear?
Authors’ response: Revised as leaf [Line No. 102].
Results
Authors’ response:
- Show the key results supporting your main findings only. Some descriptions are not clears, such as L329-331, citing 10 percentages together for two crops and five elements is too complicated. You may talk about the increase trends and increment ranges.
Authors’ response: Agreed. An increment range of treated plants and control plants is now mentioned [Line No. 280].
- L231-252 This information was missing from Methods section. Need to remove and add to Method.
Authors’ response: These lines belong to 2.7.5. Estimation of Osmolyte, Sugar, Protein, and Amino Acid Contents in Plants and they are the part of materials and methods section
- From Fig.1 bar data show it even tolerated the highest concentration of all salt forms. Not much differences in the growth under 100 and 200 uM. This needs explanations in Discussion section.
Authors’ response: Fig 1 shows that the organism tolerated highest concentration (160 mM) of NaCl only. [Line No. 238].
Discussion
- It would be helpful if the discussion is presented under several subtitles to highlight the main issues to be discussed.
Authors’ response: As per the format of journal subtitles are not allowed under discussion. However, we have made several subtitles as mentioned below
4.1. Screening for Salinity Stress Tolerance
4.2. Screening and Production of Salinity Ameliorating Traits
4.3. Confirmation of non-pathogenicity of K.variicola SURYA6
4.4. Plant Growth Promotion Studies - Pot Assay
4.5. Analysis of Osmolytes and Biochemical Contents in Wheat and Maize Plants
4.6. Analysis of Mineral Content of wheat and Maize Plants
4.7. Analysis of Soil Physical Parameters and Nutrients
- “These nutrients have been reported to reduce toxic ion accumulation and enhancement of K+/Na+ ratios in plants”. Which nutrients function to avoid toxin accumulation while balancing K+/Na+? How did you come to this from the above sentence? K+/Na+ should be “K+/Na+”, please check elsewhere.
Authors’ response: Corrected as secretion of EPS has been reported to [Line No. 433-434]
K+/Na+ are now revised as K+/Na+ [Line No. 434,438,440]
Conclusion
- First two sentences can be deleted – they are not conclusion from this study.
Authors’ response: Agreed. The first two sentences of conclusion are now deleted. [Line No. 468-471]
The writing needs lots of work, including grammatical errors and logical expressions. For example:
- L23 not clear of what kind of “chemical approaches” often used.
Authors’ response: Grammatical errors have been rectified throughout the MSS. Line 23 chemical approaches etc have been deleted.
- L38 separate “seed germination” and “root length” with a coma
Authors’ response: separated
- L39 It is better to rephrase “shoot length” as “shoot height”
Authors’ response:
- L40 add coma both before and after the word “respectively”
Authors’ response: Coma added before and after word “respectively”
- L43-44 no need to use abbreviations when use only once.
Authors’ response: These minerals are mentioned in the consequent sections hence they are abbreviated at their first appearance here
- L48 correct errors of “a higher concentrations”
Authors’ response: These words have been deleted during shortening of Abstract as per comment 2
- L51 why use capital W for wheat? Also L96, L100. Please check throughout the manuscript.
Authors’ response: Wheat is corrected as wheat throughout the MSS
- L104-105 Grammatical error. Insert “and” before “were procured from…”
Authors’ response: Word “and” added as per the suggestion [Line No. 90]
- Mixture that did not contain…,rather than “contained”. [Line No. 102]
Authors’ response: Corrected as per the suggestion
- L436 “ratios” rather than “ratio”
Authors’ response: Corrected as per the suggestion [Line No. 434]
- L439 “K+/Na+” should be “K+/Na+ratio” instead?
Authors’ response: Corrected as per the suggestion [Line No. 434,438,440]
- L439,452,454. Change “enhances”, ”accumulates”, and ”helps” into their plural form, respectively.
Authors’ response: Revised as per the suggestion [Line No. 421,452,455]

Round 2
Reviewer 1 Report
(1) The randomized complete block design (RCBD) is chosen to control experimental variation due to spatial effects. An ANOVA model for RCBD should be: Response = Treatment + Rep + Treatment X Rep. Please note that Rep is considered as block here for RCBD.
But the authors used the ANOVA model as: Response = Treatment. Here the effect of block i.e., replication is not considered. The ANOVA model used by the authors is applicable to completely randomized design (CRD).
If it were a field research, I cannot recommend changing experimental design. The only possibility in case of the field research can be the re-analysis of data as per the RCBD. For the greenhouse pot experiment, I assume there should not be a substantial difference in the results by the CRD and RCBD experimental designs.
Therefore, there are two options to correct the statistical analysis for pot experiment: (i) choose correct experimental design as per the analysis or (ii) re-analyze the data as per the experimental design chosen.
(2) How the ANOVA was conducted is not clear from the manuscript. Please mention the software used for ANOVA. For example, please write something like “The data on ____ were analyzed using SAS version 9.4 (SAS Institute, 2013)” or “The data on ____ were analyzed using Excel”, “ ANOVA was performed using the GLM procedure of SAS”____. If some of the data were analyzed by one method and the other data by another method, please mention all.
(3) Please replace “Turkey’s HSD test” with “Tukey’s HSD test” throughout the entire manuscript. Please make sure there should not be “r” in the word “Tukey”.
(4) In Figure 3B bar diagrams, the color is same for T1 and T3. Please use different colors if possible.
Author Response
Response to Reviewer 1 Round 2 Report
Comments and Suggestions for Authors
- The randomized complete block design (RCBD) is chosen to control experimental variation due to spatial effects. An ANOVA model for RCBD should be: Response = Treatment + Rep + Treatment X Rep. Please note that Rep is considered as block here for RCBD.
But the authors used the ANOVA model as: Response = Treatment. Here the effect of block i.e., replication is not considered. The ANOVA model used by the authors is applicable to completely randomized design (CRD).
If it were a field research, I cannot recommend changing experimental design. The only possibility in case of the field research can be the re-analysis of data as per the RCBD. For the greenhouse pot experiment, I assume there should not be a substantial difference in the results by the CRD and RCBD experimental designs.
Therefore, there are two options to correct the statistical analysis for pot experiment: (i) choose correct experimental design as per the analysis or (ii) re-analyze the data as per the experimental design chosen.
Authors response: We are thankful to the reviewer for pointing this discrepancy. This research included a pot study, so CRD (completely randomized design) was used. Somehow, by mistake, RCBD was mentioned. As per the suggestion of the worthy reviewer, re-analysis of all data has been carried by ANOVA models for CRD through Statistix 8.1.
- How the ANOVA was conducted is not clear from the manuscript. Please mention the software used for ANOVA. For example, please write something like “The data on ____ were analyzed using SAS version 9.4 (SAS Institute, 2013)” or “The data on ____ were analyzed using Excel”, “ ANOVA was performed using the GLM procedure of SAS”____. If some of the data were analyzed by one method and the other data by another method, please mention all.
Authors response: ANOVA analysis has been done through Statistix 8.1.
- Please replace “Turkey’s HSD test” with “Tukey’s HSD test” throughout the entire manuscript. Please make sure there should not be “r” in the word “Tukey”.
Authors response: Thanks for your minute observation. The word Turkey’s is corrected as Tukey’s
- In Figure 3B bar diagrams, the color is the same for T1 and T3. Please use different colors if possible.
Authors response: In Figure 3B bar diagrams, different colors for T1 and T3 are now used

Reviewer 2 Report
The authors have carefully revised the manuscript according to my comments. The revised manuscript has been largely improved. There are a few more concerns of the manuscript requiring minor revisions:
Introduction: Please provide some examples showing the changes in osmolytes like proline and sugars and also nutrients in crops following PGPR inoculation under control and saline conditions.
Materials and Methods: What was the Na content in soil by the end of the experiment? To measure root length, how the seedling roots were uprooted without damage to their intactness on the whole. Section 2.75 - Specify which part of plant material, leaves or roots, or the whole seedlings, were used for measuring osmolyte etc.
Results: Line.285-286: Not clear if the comparison present referred to 30 DAS or 45 DAS. Line.302-303: What the percent increment presented was compared to bacterized or non-bacterized plant? Not clear. Again the comparison referred to 30 DAS or 45 DAS, and wheat or maize plants - not clear. Line.341 “reduced the electrical conductivity of soil from 5.7 to 4.0 and pH from alkalinity (9.2) to neutrality (6.8)”, the shown data are not reflected in Table 3. Please check.
Discussion: According to your results, the promoted germination or growth promotion by IAA produced by K. variicola is supposed to be supported by or in line with those studies conducted by other researchers. An increase in chlorophyll content translated into higher photosynthesis and better growth. This needs to add references supporting your claim. Line.426-431: This part is quite general and repeated on the positive role of EPS and ACCD produced by PGPR - needs rewrite. Several places showed EPS-producing PGPR leading to the increased tissue K+/Na+ ratio, but tissue K+/Na+ ratio was not shown in this study.
Author Response
Reviewer 2 Round 2 Report
Comments and Suggestions for Authors
The authors have carefully revised the manuscript according to my comments. The revised manuscript has been largely improved. There are a few more concerns of the manuscript requiring minor revisions:
- Introduction:Please provide some examples showing the changes in osmolytes like proline and sugars and also nutrients in crops following PGPR inoculation under control and saline conditions.
Authors response: Three examples showing the changes in osmolytes like proline and sugars and also nutrients in rice and wheat crops following inoculation of Enterobacter sp. E. clacae and Arthrobacter sp. under control and saline conditions cited in Line 68
- Materials and Methods: What was the Na content in soil by the end of the experiment? To measure root length, how the seedling roots were uprooted without damage to their intactness on the whole. Section 2.75 - Specify which part of plant material, leaves or roots, or the whole seedlings, were used for measuring osmolyte etc.
Authors response: At the end of the experiment the observed level of NaCl was 100 mM NaCl/Kg soil. [Line 187-188]
Section 2.75 – Seedling part of plant material was used for measuring osmolyte etc. It is already mentioned at Line No. 211
- Results: 285-286: Not clear if the comparison present referred to 30 DAS or 45 DAS. Line.302-303: What the percent increment presented was compared to bacterized or non-bacterized plant? Not clear. Again the comparison referred to 30 DAS or 45 DAS, and wheat or maize plants - not clear. Line.341 “reduced the electrical conductivity of soil from 5.7 to 4.0 and pH from alkalinity (9.2) to neutrality (6.8)”, the shown data are not reflected in Table 3. Please check.
Authors response:
Line.285-286: The comparison referred to both 30 DAS as well as 45 DAS. Mentioned in Line 285-287
Line.302-303: A 52.0% (Figure 5C), and 309.9% (Figure 5D), 245.5% (Figure 5A), 78.3% (Figure 5B), 86.0% (Figure 5C) and 146.4% (Figure 5D) increase proline, sugars, protein and amino acid contents in wheat and maize test seedlings at 45 DAS was evident over the control. [Line No.299-301]
Line.341: Values in Table 3 are corrected now.
- Discussion: According to your results, the promoted germination or growth promotion by IAA produced by K. variicola is supposed to be supported by or in line with those studies conducted by other researchers.
Authors response: Plant growth promotion studies by IAA producing K.pneumonae and Klebsiella sp. SBP-8 is now cited [Line No.383-387]
- An increase in chlorophyll content translated into higher photosynthesis and better growth. This needs to add references supporting your claim. Line.426-431: This part is quite general and repeated on the positive role of EPS and ACCD produced by PGPR - needs rewrite. Several places showed EPS-producing PGPR leading to the increased tissue K+/Na+ratio, but tissue K+/Na+ ratio was not shown in this study.
Authors response: Line.426-431- Are deleted as suggested
